# Calibrated Reliable Regression using Maximum Mean Discrepancy

**Peng Cui[1][2],     Wenbo Hu[1][2],     Jun Zhu[1]***

[1] Dept. of Comp. Sci. & Tech., Institute for AI, BNRist Center
Tsinghua-Bosch Joint ML Center, THBI Lab, Tsinghua University, Beijing, 100084 China
[2] RealAI
xpeng.cui@gmail.com, wenbo.hu@realai.ai, dcszj@tsinghua.edu.cn

## Abstract

Accurate quantification of uncertainty is crucial for real-world applications of machine learning. However, modern deep neural networks still produce unreliable predictive uncertainty, often yielding over-confident predictions. In this paper, we are concerned with getting well-calibrated predictions in regression tasks. We propose the calibrated regression method using the maximum mean discrepancy by minimizing the kernel embedding measure. Theoretically, the calibration error of our method asymptotically converges to zero when the sample size is large enough. Experiments on non-trivial real datasets show that our method can produce well-calibrated and sharp prediction intervals, which outperforms the related state-of-the-art methods.

## 1   Introduction

Deep learning has achieved significant progress on a wide range of complex tasks [22] mainly in terms of some metrics on prediction accuracy. However, high accuracy alone is often not sufficient to characterize the performance in real applications, where uncertainty is pervasive because of various facts such as incomplete knowledge, ambiguities, and contradictions. Accurate quantification of uncertainty is crucial to derive a robust prediction rule. For example, an accurate uncertainty estimate can reduce the occurrence of accidents in medical diagnosis [23], warn users in time in self-driving systems [27], reject low-confidence predictions [6], and better meet consumers' order needs of internet services especially on special events [39]. In general, there are two main types of uncertainty, *aleatoric* uncertainty and *epistemic* uncertainty [10]. *Aleatoric* uncertainty captures inherent data noise (e.g., sensor noise), while *epistemic* uncertainty is considered to be caused by model parameters and structure, which can be reduced by providing enough data.

Though important, it is highly nontrivial to properly characterize uncertainty. Deep neural networks (DNNs) typically produce point estimates of parameters and predictions, and are insufficient to characterize uncertainty because of their deterministic functions [11]. It has been widely observed that the modern neural networks are not properly calibrated and often tend to produce over-confident predictions [3, 15]. An effective uncertainty estimation is to directly model the predictive distribution with the observed data in a Bayesian style [26]. But performing Bayesian inference on deep networks is still a very challenging task, where the networks define highly nonlinear functions and are often over-parameterized [38, 34]. The uncertainty estimates of Bayesian neural networks (BNNs) may lead to an inaccurate uncertainty quantification because of either model misspecification or the use of approximate inference [19]. Besides, BNNs are computationally more expensive and slower to train in practice, compared to non-Bayesian NNs. For example, a simple method of MC-Dropout

directly captures uncertainty without changing the network structure [12]. But the uncertainty quantification of MC-Dropout can be inaccurate, as will be seen in the empirical results in this paper.

Apart from BNNs, some methods have been developed to incorporate the variance term into NNs to estimate the predictive uncertainty. For instance, [17] proposed a heteroscedastic neural network (HNN) to combine both model uncertainty and data uncertainty simultaneously, getting the mean and variance by designing two outputs in the last layer of the network. Based on HNN, [21] described a simple and scalable method for estimating predictive uncertainty from ensembled HNNs, named as *deep ensemble*. But the ensembled model is usually computationally expensive especially when the model structure is complex.

An alternative way to obtain accurate predictive uncertainty is to calibrate the inaccurate uncertainties. Early attempts were made to use the scaling and isotonic regression techniques to calibrate the supervised learning predictions of traditional models, such as SVMs, neural networks and decision trees [32, 29]. For regression tasks, the prediction intervals are calibrated based on the proportion of covering ground truths. Recently, [15, 19] adopted a post-processing step to adjust the output probabilities of the modern neural networks based on the temperature scaling and non-parametric isotonic regression techniques. Such *post-processing* methods can be directly applied to both BNNs and DNNs without model modifications. But they need to train an auxiliary model and rely on an additional validation dataset. Moreover, the isotonic regression tends to overfit especially for small datasets [35]. [31, 36] directly incorporated a calibration error to loss functions to obtain the calibrated prediction intervals at the specific confidence level. Predetermining the specific confidence level can be regarded as the "point calibration" and its calibration model needs to be retrained when the confidence level is changed. [35] proposed an extension to the post-processing procedure of the isotonic regression, using Gaussian Processes (GPs) and Beta link functions. This method improves calibration at the distribution level compared to existing post-processing methods, but is computationally expensive because of the GPs.

In this paper, we propose a new way to obtain the calibrated predictive uncertainty of regression tasks at the global quantile level — it derives a distribution matching strategy and gets the well-calibrated distribution which can output predictive uncertainties at all confidence levels. Specifically, we minimize the maximum mean discrepancy (MMD) [13] to reduce the distance between the predicted probability uncertainty and true one. We show that the calibration error of our model asymptotically converges to zero when the sample size is sufficiently large. Extensive empirical results on the regression and time-series forecasting tasks show the effectiveness and flexibility of our method.

## 2 Preliminaries

In this section, we introduce some preliminary knowledge of calibrated regressor and maximum mean discrepancy, as well as the notations used in the sequel.

### 2.1 Calibrated Regressor

Let us denote a predictive regression model as $f\colon x \to y$, where $x \in \mathbb{R}^d$ and $y \in \mathbb{R}$ are random variables. We use $\Theta$ to denote the parameters of $f$. We learn a proposed regression model given a labeled dataset $\{(x_i, y_i)\}_{i=1}^{N}$ with $N$ samples.

To obtain more detailed uncertainty of the output distribution, a calibrated regressor outputs the cumulative distribution function (CDF) $F_i$ by the predictive distribution for each input $x_i$. When evaluating the calibration of regressors, the inverse function of CDF $F_i^{-1} : [0, 1] \to \hat{y_i}$ is used to denote the quantile function:

$$F_i^{-1}(p) = \inf \left\{ y : p \leq F_i(y) \right\}. \tag{1}$$

Intuitively, the calibrated regressor should produce calibrated prediction intervals (PIs). For example, given the probability $95\%$, the calibrated regressor should output the prediction interval that approximately covers $95\%$ of ground truths in the long run.

Formally, we define a *well-calibrated* regressor [8, 19] if the following condition holds, for all $p \in [0, 1]$,

$$\frac{\sum_{i=1}^{N} \mathbb{I}\left\{y_i \leq F_i^{-1}(p)\right\}}{N} \to p, \text{when } N \to \infty, \tag{2}$$

where $\mathbb{I}(\cdot)$ is the indicator function that equals to 1 if the predicate holds otherwise 0. More generally, for a prediction interval $[F_i^{-1}(p_1), F_i^{-1}(p_2)]$, there is a similar definition of two-sided calibration as follows:

$$\frac{\sum_{i=1}^{N} \mathbb{I}\left\{F_i^{-1}(p_1) \leq y_i \leq F_i^{-1}(p_2)\right\}}{N} \to p_2 - p_1 \quad \text{for all } p_1, p_2 \in [0, 1] \tag{3}$$

as $N \to \infty$.

For this task, the previous methods applied the post-processing techniques [19, 15] or added a regularized loss [31, 36]. But when we want to get PIs with different confidence levels, we need to retrain the model because the confidence level is predetermined in the loss function. In contrast, we argue that the key challenge for the calibrated regression is getting the well-calibrated distribution. Based on this principle, our method utilizes the distribution matching strategy and aims to directly get a calibrated predictive distribution, which can naturally output well-calibrated CDF and PIs for each input $x_i$.

## 2.2 Maximum Mean Discrepancy

Our method adopts maximum mean discrepancy (MMD) to perform distribution matching. Specifically, MMD is defined via the Hilbert space embedding of distributions, known as kernel mean embedding [13]. Formally, given a probability distribution, the kernel mean embedding represents it as an element in a reproducing kernel Hilbert space (RKHS). An RKHS $\mathcal{F}$ on $\mathcal{X}$ with the kernel function $k$ is a Hilbert space of functions $g : \mathcal{X} \to \mathbb{R}$. We use $\phi(x) = k(x, \cdot)$ to represent the feature map of $x$. The expectation of embedding on its feature map is defined as:

$$\mu_X := \mathbb{E}_X[\phi(X)] = \int_\Omega \phi(x) P(dx). \tag{4}$$

This kernel mean embedding can be used for density estimation and two-sample test [13].

Based on the Hilbert space embedding, the maximum mean discrepancy (MMD) estimator was developed to distinguish two distributions $P$ and $Q$ [13]. Formally, the MMD measure is defined as follows:

$$L_m(P, Q) = \|\mathbb{E}_X(\phi(P)) - \mathbb{E}_X(\phi(Q))\|_{\mathcal{F}}. \tag{5}$$

The MMD estimator is guaranteed to be unbiased and has nearly minimal variance among unbiased estimators [25]. Moreover, it was shown that $L_m(P, Q) = 0$ if and only if $P = Q$ [13].

We conduct a hypothesis test with null hypotheses $H_0 : P = Q$, and the alternative hypotheses $H_1 : P \neq Q$ if $L_m(P, Q) > c_\alpha$ for some chosen threshold $c_\alpha > 0$. With a characteristic kernel function (e.g., the popular RBF kernels), the MMD measure can be used to distinguish the two different distributions and have been applied to generative modeling [24, 25].

In practice, the MMD objective can be estimated using the empirical kernel mean embeddings:

$$\hat{L}_m^2(P, Q) = \left\| \frac{1}{N} \sum_{i=1}^{N} \phi(x_{1i}) - \frac{1}{M} \sum_{j=1}^{M} \phi(x_{2j}) \right\|_{\mathcal{F}}^2, \tag{6}$$

where $x_{1i}$ and $x_{2j}$ are independent random samples drawn from the distributions $P$ and $Q$ respectively.

## 3 Calibrated Regression with Maximum Mean Discrepancy

We now present the uncertainty calibration with the maximum mean discrepancy and then plug it into the proposed calibrated regression model. We also give the theoretical guarantee to show the effectiveness of our uncertainty calibration strategy.

### 3.1 Uncertainty Calibration with Distribution Matching

Specifically in this part, we use $P$ and $Q$ to represent the unknown true distribution and predictive distribution of our regression model respectively. The distribution matching strategy of our uncertainty calibration model is to directly minimize the kernel embedding measure defined by MMD in

Eqn. (6). The specific goal is to let the predictive distribution $Q$ converge asymptotically to the unknown target distribution $P$ so that we can get the calibrated CDFs $\{F_i\}$. The strategy is to minimize the MMD distance measure between the regression ground-truth targets $\{y_1, \cdots, y_n\}$ and random samples $\{\hat{y}_1, \cdots, \hat{y}_n\}$ from the predictive distribution $Q$. The specific form of the MMD distance loss $L_m$ is:

$$L_m^2(P, Q) := \left\| \frac{1}{N} \sum_{i=1}^{N} \phi(y_i) - \frac{1}{N} \sum_{j=1}^{N} \phi(\hat{y}_j) \right\|_{\mathcal{F}}^2. \tag{7}$$

We use a mixture of $k$ kernels spanning multiple ranges for our experiments:

$$k(x, x') = \sum_{i=1}^{K} k_{\sigma_i}(x, x'), \tag{8}$$

where $k_{\sigma_i}$ is an RBF kernel and the bandwith parameter $\sigma_i$ can be chosen simple values such as 2,4,8, etc. The kernel was proved to be characteristic, and it can maximize the two-sample test power and low test error [14]. In general, a mixture of five kernels or more is sufficient to obtain good results.

With the incorporation of this MMD loss, we learn the calibrated predictive probability distribution and the obtained uncertainties can be generalized to arbitrary confidence levels without retraining.

In theory, under $H_0 : P = Q$, the predictive distribution $Q$ will converge asymptotically to the true distribution $P$ as sample size $N \to \infty$, which is why minimizing MMD loss is effective for uncertainty calibration. Leveraging our distribution matching strategy, the uncertainty calibration can be achieved by narrowing the gap between $P$ and $Q$. Formally, we have the following theoretical result:

**Theorem 1.** *Suppose that the predictive distribution $Q$ has the sufficient ability to approximate the true unknown distribution $P$, given data is i.i.d. Eqn. (9) holds by minimizing the MMD loss $L_m = \|\mu_{x_1} - \mu_{x_2}\|_{\mathcal{F}}$ in our proposed methodology as the sample size $N \to \infty$*

$$\frac{\sum_{i=1}^{N} \mathbb{I}\left\{ y_i \leq F_i^{-1}(p) \right\}}{N} \to p \quad \textit{for all } p \in [0, 1] \tag{9}$$

*Proof.* $L_m(P, Q) = 0$ if and only if $P = Q$ when $\mathcal{F}$ is a unit ball in a universal RKHS [13]. Under $H_0 : P = Q$, the predictive distribution $Q(x)$ will converge asymptotically to the unknown true distribution $P(x)$ as the sample size $N \to \infty$ by minimizing the MMD loss $L_m$. Further, Eqn. (9) holds according to the obtained predictive distribution. Because the confidence level $p$ is exactly equal to the proportion of samples $\{y_1, \cdots, y_n\}$ covered by the prediction interval. $\square$

This theoretical result can be generalized to the two side calibration condition defined in Eqn. (3) and we defer the details to Appendix A.

## 3.2 Calibrated Regression with MMD

To represent the model uncertainty, we use a heteroscedastic neural network (HNN) to get the predictive distribution and outputs the predicted mean $\mu(x)$ and the variance $\sigma^2(x)$ in the final layer, which can combine epistemic uncertainty and aleatoric uncertainty in one model [30, 17].

Based on this representation model, we use a two-stage learning framework which optimizes the two objectives one by one, namely the negative log likelihood loss $L_h$ and the uncertainty calibration loss $L_m$. In the first stage, the optimal model parameters can be learned by minimizing negative log-likelihood loss (NLL):

$$L_h(\Theta) = \sum_{i=1}^{N} \frac{\log \sigma_\Theta^2(x_i)}{2} + \frac{(y_i - \mu_\Theta(x_i))^2}{2\sigma_\Theta^2(x_i)} + \text{ constant}. \tag{10}$$

In practice, to improve numerical stability, we optimize the following equivalent form:

$$L_h(\Theta) = \sum_{i=1}^{N} \frac{1}{2} \exp(-s_i)(y_i - \mu_\Theta(x_i))^2 + \frac{1}{2} s_i, \quad s_i := \log \sigma_\Theta^2(x_i). \tag{11}$$

Although the Gaussian assumption is a bit restrictive above, we found that the method performs satisfactorily well in our experiments. In the second stage, we minimize the uncertainty calibration loss with MMD, i.e., Eqn. (7).

The overall procedure is two-stage:

$$
\begin{array}{ll}
\text{step 1:} & \min_\Theta L_h(\Theta; y, f(x)), \\
\text{step 2:} & \min_\Theta L_m(\Theta; y, f(x)),
\end{array}
\tag{12}
$$

where $L_m$ is the loss function of distribution matching objective, and $L_h$ is the loss function of distribution estimation. We detail the whole process of the framework in Algorithm 1. The main merits of the two-stage learning are to 1) utilize the representation capability of the HNN model in the first stage and 2) learn the calibrated predictive distribution via the distribution matching strategy in the second stage. Compared with the bi-level learning algorithm used in [19] which iterates the two stages for several times, our method runs with one time iteration of the two stages, which reduces the computation cost of the kernel-based MMD component.

**Comparison with Post-processing Calibration Methods** The previous post-processing methods [19, 36, 31] calibrate the uncertainty outputs of the input dataset without any model modifications and needs to be retrained when the confidence level is changed. In contrast, the proposed distribution matching with MMD, albeit also regarded as a post-processing procedure, learns the calibrated predictive model, which means practitioners are not required to retrain the model but can enjoy the calibration performance.

---

**Algorithm 1** Deep calibrated reliable regression model.

---

**Input:**
    Labeled training data and kernel bandwidth parameters
**Output:**
    Trained mean $\mu(x_i)$ and variance $\sigma(x_i)$ for the predictive distribution
 1: **while** not converged **do**
 2:    Compute $\mu(x_i)$ and $\log \sigma(x_i)$
 3:    Compute NLL loss $L_h$ by Eqn. (11)
 4:    Update model parameters $\Theta = \arg\min_\Theta \mathrm{L}_h(\Theta; y, f(x))$ by SGD
 5: **end while**
 6: **while** not converged **do**
 7:    Compute $\mu(x_i)$ and $\log \sigma(x_i)$, randomly sampling data $\{\hat{y}_i\}_{i=1}^N$ from predictive distrbution
 8:    Compute MMD loss $L_m$ by Eqn. (7)
 9:    Update model parameters $\Theta = \arg\min_\Theta \mathrm{L}_m(\Theta; y, f(x))$ by SGD
10: **end while**
11: **return** a trained model $f(x)$;

---

## 4 Experiments

In this section, we compare the proposed method with several strong baselines on the regression and time-series forecasting tasks in terms of predictive uncertainty. The time-series forecasting task models multiple regression sub-problems in sequence and the tendency along the sliding windows can be used to examine the obtained predictive uncertainty. Then we show the sensitivity analysis and the time efficiency of our proposed method.

### 4.1 Datasets and Experimental Settings

**Baselines** We compare with several competitive baselines, including MC-Dropout (MCD) [12], Heteroscedastic Neural Network (HNN) [17], Deep Ensembles (Deep-ens) [21], Ensembled Likehood (ELL), MC-Dropout Likelihood (MC NLL), Deep Gaussian Processes (DGP) [33] and the post-hoc calibration method using isotonic regression (ISR) [19]. ELL and MC NLL are our proposed variants inspired by Deep Ensemble. The variance of ELL is computed by predictions from multiple networks during the training phase, and the variance of MC NLL is computed by multiple random predictions based on MC-Dropout during the training phase. Details of these compared methods can be found in Appendix B.1.

**Hyperparameters** For all experimental results, we report the averaged results and std. errors obtained from 5 random trials. The details of hyperparameters setting can be found in Appendix B.2.

**Datasets** We use several public datasets from UCI repository [2] and Kaggle [1]: 1) for the time-series task: Pickups, Bike-sharing, PM2.5, Metro-traffic and Quality; 2) for the regression task: Power Plant, Protein Structure, Naval Propulsion and wine. The details of the datasets can be found in Appendix B.3.

**Evaluation Metrics** We evaluate the performance using two metrics: 1) RMSE for the prediction precision; and 2) the calibration error. The calibration error is the absolute difference between true confidence and empirical coverage probability. We use two variants: the expectation of coverage probability error (ECPE) and the maximum value of coverage probability error (MCPE). We put the detailed definitions of the metrics in Appendix C.

## 4.2 Results of Time-series Forecasting Tasks

For time-series forecasting tasks, we construct an LSTM model with two hidden layers (128 hidden units and 64 units respectively) and a linear layer for making the final predictions. The size of the sliding window is 5 and the forecasting horizon is 1. Take the Bike-sharing dataset as an example, the bike sharing data of the past five hours will be used to predict the data of one hour in the future. All datasets are split into 70% training data and 30% test data.

| Dataset | Metric | MCD | HNN | Deep-ens | MC NLL |
|---|---|---|---|---|---|
| Metro-traffic | ECPE | 0.304±0.005 | 0.102±0.002 | 0.100±0.001 | 0.142±0.010 |
| | MCPE | 0.505±0.011 | 0.162±0.003 | 0.160±0.002 | 0.235±0.014 |
| | RMSE | 523.6±6.725 | 556.3±3.332 | **508.9±1.288** | 631.6±14.23 |
| Bike-sharing | ECPE | 0.258±0.011 | 0.054±0.002 | 0.038±0.001 | 0.119±0.013 |
| | MCPE | 0.432±0.020 | 0.089±0.004 | 0.066±0.008 | 0.206±0.022 |
| | RMSE | 38.86±0.141 | 40.71±0.542 | **37.60±0.355** | 61.57±1.624 |
| Pickups | ECPE | 0.246±0.017 | 0.078±0.001 | 0.064±0.006 | 0.088±0.016 |
| | MCPE | 0.408±0.025 | 0.117±0.005 | 0.098±0.011 | 0.136±0.010 |
| | RMSE | 350.3±6.562 | 359.8±3.421 | 336.4±1.653 | 526.8±9.214 |
| PM2.5 | ECPE | 0.331±0.013 | 0.022±0.001 | 0.026±0.003 | 0.081±0.010 |
| | MCPE | 0.550±0.025 | 0.050±0.004 | 0.060±0.004 | 0.151±0.027 |
| | RMSE | 70.95±2.629 | 58.81±0.372 | 60.24±0.114 | 66.77±3.613 |
| Air-quality | ECPE | 0.329±0.005 | 0.058±0.003 | 0.045±0.001 | 0.111±0.004 |
| | MCPE | 0.561±0.008 | 0.091±0.006 | 0.072±0.002 | 0.178±0.004 |
| | RMSE | 81.16±0.111 | **79.60±0.254** | 80.03±0.236 | 87.12±0.971 |

| Dataset | Metric | ELL | DGP | ISR | proposed |
|---|---|---|---|---|---|
| Metro-traffic | ECPE | 0.048±0.017 | 0.115±0.007 | 0.032±0.002 | **0.017±0.001** |
| | MCPE | 0.075±0.027 | 0.192±0.013 | 0.051±0.003 | **0.036±0.002** |
| | RMSE | 613.5±18.63 | 646.4±0.302 | 556.3±3.332 | 545.5±4.225 |
| Bike-sharing | ECPE | 0.027±0.003 | 0.121±0.003 | 0.042±0.002 | **0.006±0.001** |
| | MCPE | 0.048±0.055 | 0.213±0.005 | 0.066±0.005 | **0.019±0.002** |
| | RMSE | 52.50±2.901 | 55.39±0.397 | 40.71±0.542 | 37.93±0.334 |
| Pickups | ECPE | 0.018±0.008 | 0.098±0.003 | 0.049±0.002 | **0.008±0.001** |
| | MCPE | 0.038±0.016 | 0.160±0.005 | 0.075±0.004 | **0.023±0.001** |
| | RMSE | **325.9±11.23** | 440.3±3.469 | 359.8±3.421 | 346.9±4.652 |
| PM2.5 | ECPE | 0.080±0.007 | 0.061±0.006 | 0.023±0.002 | **0.010±0.000** |
| | MCPE | 0.119±0.011 | 0.149±0.014 | 0.057±0.006 | **0.035±0.003** |
| | RMSE | 61.09±0.434 | 61.44±2.113 | 58.81±0.372 | **57.43±0.332** |
| Air-quality | ECPE | 0.018±0.005 | 0.102±0.002 | 0.030±0.001 | **0.010±0.001** |
| | MCPE | 0.04±0.008 | 0.181±0.003 | 0.044±0.005 | **0.026±0.001** |
| | RMSE | 90.01±0.566 | 86.05±0.210 | 79.60±0.254 | 80.69±0.292 |

Table 1: The forecast and calibration error scores of each method on different datasets. Each row corresponds to the results of a specific method in a particular metric.

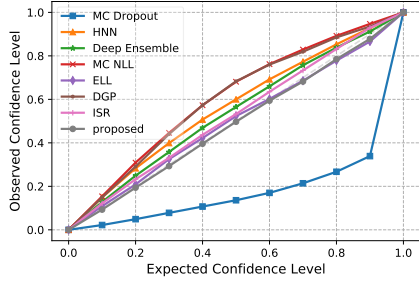 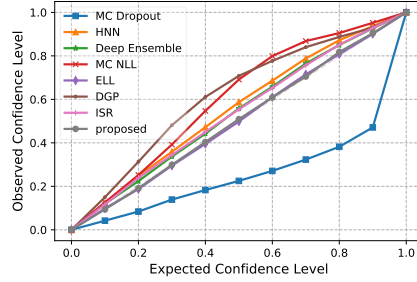

(a) Dataset: Air Quality.　　　　　　(b) Dataset: Bike Sharing.

Figure 1: For the time-series forecasting task, we plot the expected confidence vs observed confidence for all methods. The closer to the diagonal line, the uncertainty calibration is better. The results of other datasets can be found in Appendix.

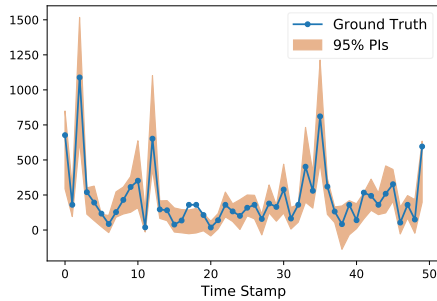 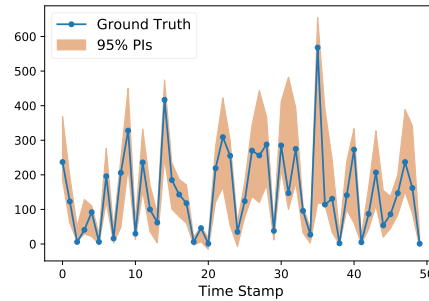

(a) Dataset: Air Quality.　　　　　　(b) Dataset: Bike Sharing.

Figure 2: Calibrated forecasting: Displayed prediction intervals (PIs) obtained at the 95% confidence level by our proposed method in a time-series. As shown in the figure, the prediction intervals are also sharp while accurately covering the ground truth. The results of other datasets can be found in Appendix.

Table 1 present the results of all the methods, including the forecast and calibration errors. We can see that our method with the MMD distribution matching strategy achieves the accurate forecasting results on par with the strong baselines in terms of RMSE[2], but significantly outperforms the baselines in the uncertainty calibration, in terms of ECPE and MCPE on all data-sets. Besides, we prefer prediction intervals as tight as possible while accurately covering the ground truth in regression tasks. We measure the sharpness using the width of prediction intervals, which is detailed in Appendix C. And our method also gets a relatively tighter prediction interval through the reported calibration sharpness from Table 4 in the Appendix. In addition, the ensemble method is second only to ours, due to the powerful ability of the ensemble of multiple networks. But when the network complexity is greater than the data complexity, the computation of the ensemble method is quite expensive, while our method can also be applied to more complex NNs. Figure 1 shows the proportion that PIs covering ground truths at different confidence levels. The result of our model is closest to the diagonal line, which indicates the best uncertainty calibration among all methods. Figure 2 shows the predictions and corresponding 95% prediction intervals. The intervals are visually sharp and accurately cover the ground truths.

### 4.3　Results of Regression Tasks

For regression tasks, we used a fully connected neural network with two hidden layers (256 hidden units) and each layer has a ReLU activation function. The size of our networks is close to the previous works [12, 19, 21] on regression problems. We evaluate on four UCI datasets varying in size from 4,898 to 45,730 samples. We randomly split 80% of each data-set for training and the rest for testing. Table 2 presents the results of all methods, where we can draw similar conclusions as in the time-series forecasting tasks. The forecast results of our method is competitive in terms of RMSE and the calibration error of our method is significantly smaller than existing methods. Figure

3 reflects that the uncertainty calibration performance of each method at different confidence levels in general regression tasks and we find that our method significantly improves calibration.

| Dataset | Metric | MCD | HNN | Deep-ens | MC NLL |
|---|---|---|---|---|---|
| Power Plant | ECPE | 0.235±0.021 | 0.094±0.002 | 0.084±0.004 | 0.095±0.004 |
| | MCPE | 0.386±0.038 | 0.151±0.007 | 0.142±0.001 | 0.153±0.004 |
| | RMSE | **3.792±0.171** | 3.843±0.165 | 3.945±0.150 | 3.936±0.158 |
| Protein Structure | ECPE | 0.365±0.011 | 0.042±0.006 | 0.049±0.001 | 0.086±0.005 |
| | MCPE | 0.635±0.021 | 0.071±0.005 | 0.084±0.002 | 0.138±0.005 |
| | RMSE | **4.088±0.014** | 4.337±0.021 | 4.255±0.010 | 4.574±0.018 |
| Naval Propulsion | ECPE | 0.175±0.077 | 0.038±0.006 | 0.270±0.016 | 0.216±0.042 |
| | MCPE | 0.283±0.116 | 0.065±0.007 | 0.431±0.025 | 0.344±0.047 |
| | RMSE | 0.001±0.000 | 0.001±0.000 | 0001±0.000 | 0.001±0.001 |
| Wine | ECPE | 0.235±0.021 | 0.041±0.003 | 0.012±0.001 | 0.046±0.006 |
| | MCPE | 0.386±0.038 | 0.082±0.013 | 0.034±0.004 | 0.095±0.011 |
| | RMSE | 0.732±0.041 | 0.705±0.038 | **0.672±0.040** | 0.683±0.064 |

| Dataset | Metric | ELL | DGP | ISR | proposed |
|---|---|---|---|---|---|
| Power Plant | ECPE | 0.019±0.025 | 0.094±0.005 | 0.062±0.003 | **0.007±0.001** |
| | MCPE | 0.035±0.037 | 0.158±0.008 | 0.105±0.003 | **0.024±0.003** |
| | RMSE | 4.186±0.184 | 4.181±0.009 | 3.843±0.165 | 3.819±0.112 |
| Protein Structure | ECPE | 0.038±0.009 | 0.020±0.002 | 0.014±0.006 | **0.006±0.000** |
| | MCPE | 0.075±0.016 | 0.036±0.004 | 0.027±0.010 | **0.024±0.002** |
| | RMSE | 4.519±0.019 | 4.950±0.011 | 4.337±0.021 | 4.556±0.012 |
| Naval Propulsion | ECPE | 0.059±0.034 | 0.115±0.007 | 0.021±0.003 | **0.012±0.001** |
| | MCPE | 0.117±0.051 | 0.192±0.012 | 0.036±0.010 | **0.030±0.004** |
| | RMSE | 0.002±0.001 | 0.001±0.000 | 0.001±0.000 | **0.001±0.000** |
| Wine | ECPE | 0.073±0.009 | 0.178±0.003 | 0.083±0.006 | **0.008±0.002** |
| | MCPE | 0.103±0.011 | 0.300±0.006 | 0.127±0.008 | **0.024±0.004** |
| | RMSE | 0.684±0.061 | 0.754±0.031 | 0.705±0.038 | 0.705±0.035 |

Table 2: The calibration error scores of uncertainty evaluation and RMSE for each method on different datasets, each row has the results of a specific method in a particular metric. Our method improves calibration and outperforms all baselines

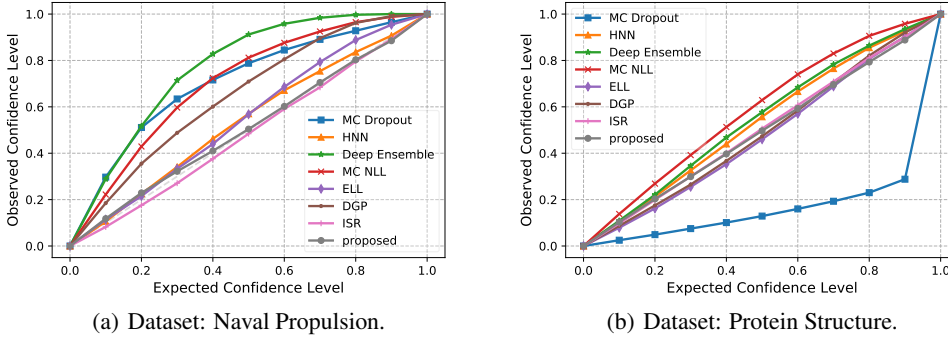

(a) Dataset: Naval Propulsion.   (b) Dataset: Protein Structure.

Figure 3: For the regression task, we plot the expected confidence vs observed confidence for all methods. The closer to the diagnoal line, the uncertainty calibration is better. The results of other datasets can be found in Appendix.

## 4.4 Computation Efficiency

We analyze the time complexity on the type of methods that generate the uncertainty distribution, these methods are relatively computationally expensive : DGP, Deep Ensembles, ELL and our proposed method. For the regression task, these four methods use the same network structure with a fully connected neural network (256 hidden units) at each hidden layer. The training and inference of DGP is performed using a doubly stochastic variational inference algorithm [33]. As can be seen

in Figure 4, DGP is the most time-consuming, the training time increases almost linearly as the number of network layers increases. The computation time of our method is the least among all methods when model complexity becomes higher, and can also keep low calibration error. This result coheres our argument that our method is not computationally expensive compared to the baseline methods.

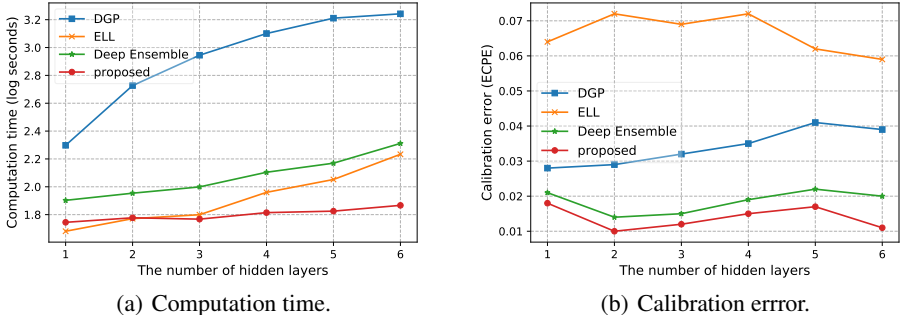

(a) Computation time.                    (b) Calibration errror.

Figure 4: The computation time (log seconds) of four methods during model training phase (left) and calibration error of different models (right) on the wine dataset on GTX1080Ti. We can see that our method is also effective in computing efficiency and calibration for more complex models.

## 5   Conclusion and Discussions

We present a flexible and effective uncertainty calibration method with the MMD distribution matching strategy for regression and time-series forecasting tasks. Our method is guaranteed to produce well-calibrated predictions given sufficient data under mild assumptions. Extensive experimental results show that our method can produce reliable predictive distributions, and obtain the well-calibrated and sharp prediction intervals.

There are several directions for future investigation. Firstly, the Gaussian likelihood may be too-restrictive sometimes and one could use a mixture distribution or a complex network, e.g., mixture density network [4] as a base model. Secondly, our calibration strategy can be extended to classification tasks. But the challenge we need to overcome is the impact of batch-size and binning on the performance of MMD. Thirdly, the kernels used in the MMD definition can be defined on other data structures, such as graphs and time-series [16]. Finally, it is interesting to investigate on the sample size for a given task. Specifically, we provide an asymptotic analysis on well-calibration, while in practice we only have finite data. [13] shows that MMD has performance guarantees at finite sample sizes, based on uniform convergence bounds. For our method, regardless of whether or not $p = q$, the empirical MMD converges in probability at rate $O((m + n)^{-\frac{1}{2}})$ to its population value, where $m$ and $n$ respectively represent the number of samples sampled from $P$ and $Q$. So a further investigation on the bound of our method is worth considering in the future work.

### Statement of Potential Broader Impact

Uncertainty exists in many aspects of our daily life, which plays a critical role in the application of modern machine learning methods. Unreliable uncertainty quantification may bring *safety and reliability* issues in these applications like medical diagnosis, autonomous driving, and demand forecasting. Despite deep learning has achieved impressive accuracies on many tasks, NNs are poor to provide accurate predictive uncertainty. Machine learning models should provide accurate confidence bounds (i.e., uncertainty estimation) on these safety-critical tasks.

This paper aims to solve the problem of inaccurate predictive quantification for regression models. Our method produces the well-calibrated predictive distribution while achieving the high-precision forecasting for regression tasks, and naturally generate reliable prediction intervals at any confidence level we need.

Our proposal has a positive impact on a variety of tasks using the regression models. For example, our proposed model produces more accurate demand forecasting based on the historical sales data for a retail company, which can calculate the safety stock to make sure you don't lose customers. We

believe that it is necessary to consider the uncertainty calibration for many machine learning models, which will improve the *safety and reliability* of machine learning and deep learning methods.

## Acknowledgement

We would like to thank the anonymous reviewers for their useful comments, especially for review 1 and review 3. Part of this work was done when the first two authors were working at RealAI. This work was supported by the National Key Research and Development Program of China (No.2017YFA0700904), NSFC Projects (Nos. 61620106010, U19B2034, U1811461), Beijing Academy of Artificial Intelligence (BAAI), Tsinghua-Huawei Joint Research Program, a grant from Tsinghua Institute for Guo Qiang, Tiangong Institute for Intelligent Computing, and the NVIDIA NVAIL Program with GPU/DGX Acceleration.

## Footnotes

*J.Z is the corresponding author.

[2]In Table 5 in the Appendix, we also show the results in other metrics, such as $R^2$, SMAPE, etc., which have a similar conclusion.

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
