[Supplementary Material]

# A  Two-sided Calibration Theorem

**Theorem 2.** *Suppose that the predictive distribution $Q$ has the sufficient ability to approximate the true unknown distribution $P$, given data is i.i.d. Eqn.* (13) *holds by minimizing the MMD loss $L_m = \|\mu_{x_1} - \mu_{x_2}\|_{\mathcal{F}}$ in our proposed methodology as the sample size $N \to \infty$*

$$\frac{\sum_{i=1}^{N} \mathbb{I}\left\{ F_i^{-1}(p_1) \le y_i \le F_i^{-1}(p_2) \right\}}{N} \to p_2 - p_1 \quad \text{for all } p_1, p_2 \in [0,1] \tag{13}$$

*Proof.* $L_m(P,Q) = 0$ if and only if $P = Q$ when $\mathcal{F}$ is a unit ball in a universal RKHS [13]. Under $H_0 : P = Q$, the predictive distribution $Q(x)$ will converge asymptotically to the unknown true distribution $P(x)$ as the sample size $N \to \infty$ by minimizing the MMD loss $L_m$. Further, Eqn. (13) holds according to the obtained predictive distribution. Because the confidence level $p_2 - p_1$ is exactly equal to the proportion of samples $\{y_1, \cdots, y_n\}$ covered by the two-sided prediction interval. $\qquad \square$

# B  Experimental Setting

## B.1  Baselines

• MC-Dropout (MCD) [12]: A variant of standard dropout, named as Monte-Carlo Dropout. Interpreting dropout in deep neural networks as approximate Bayesian inference in deep Gaussian process. Epistemic uncertainties can be quantified with a Monte-Carlo sampling sample by using dropout during the test phase in the network without changing NNs model itself. For all experiments, the dropout probability was set at 0.3. The conventional MSE loss is used in this method.

• Heteroscedastic Neural Network (HNN) [17]: In this approach, similar to a heteroscedastic regression, the network has two outputs in the last layer, corresponding to the predicted mean and variance for each input $x_i$. HNN is trained by minimizing the negative log-likelihood loss (NLL). Epistemic uncertainty and aleatoric uncertainty can be captured by using MC-Dropout.

• Deep Ensembles [21]: A simple and scalable ensembled method, here referred to as Deep-ens. In this approach, predictive uncertainty is estimated by training multiple heteroscedastic neural networks independently. Each HNN is trained with the entire training dataset. In the end, the predictive distribution is considered a uniformly-weighted Gaussian mixture. For simplicity of computation, the distribution is regarded as a Gaussian for each input $x_i$. The mean and variance of a mixture $M^{-1} \sum \mathcal{N}\left(\mu_{\Theta_m}(\mathbf{x_i}), \sigma^2_{\Theta_m}(\mathbf{x_i})\right)$ are given by $\mu_*(\mathbf{x_i}) = M^{-1} \sum_m \mu_{\Theta_m}(\mathbf{x_i})$ and $\sigma^2_*(\mathbf{x_i}) = M^{-1} \sum_m \left(\sigma^2_{\Theta_m}(\mathbf{x_i}) + \mu^2_{\Theta_m}(\mathbf{x_i})\right) - \mu^2_*(\mathbf{x_i})$ respectively, where $\{\Theta_m\}_{m=1}^{M}$ represent the parameters of the ensemble model. Hence the prediction intervals can be calculated by the CDF of Gaussian distribution.

• Ensembled Likelihood (ELL): Inspired by deep ensemble, we jointly train k networks to minimize ensembled likelihood loss (ELL) by gradient descent algorithm, e.g. SGD. Where $\mu_i = \frac{1}{k}\sum_{i=1}^{k} f_1(x_i) + f_2(x_i) + ... + f_k(x_i), \sigma_i^2 = \frac{1}{k}\sum_{i=1}^{k}(f_i(x_i) - \mu_i)^2$. Note that, there are interactions across these $k$ networks, or just look at the standard deviation which is computed across predictions from different networks. This is different from typical heteroskedastic models (trained with Eqn. (14)), where the noise std.dev. comes from a single network, and it's also different from standard deep ensembles, in which you can decompose the loss across different networks (thus the networks do not interact at all during training, training networks independently). Such lack of interaction might be wasting capacity.

$$L_h(x_i, y_i) = \sum_{i=1}^{N} \frac{1}{2} \exp\left(-s_i\right)(y_i - \mu_\Theta(x_i))^2 + \frac{1}{2}s_i, \quad s_i := \log \sigma^2_\Theta(x_i). \tag{14}$$

• MC-Dropout Likelihood (MC NLL): We designed a method for combining MC-Dropout and ensemble in a single network. The NNs perform Monte-Carlo sampling before each back-propagation during the training phase. Similar to the ELL method above, we can calculate the mean and variance of the Monte-Carlo sample. Furthermore, injecting the mean and variance into negative log-likelihood (NLL) loss to perform back-propagation.

- Deep Gaussian Processes (DGP): Deep Gaussian processes (DGPs) [7], a Bayesian inference method, are multi-layer generalizations of Gaussian processes(GPs), the data is modeled as the output of a multivariate GP, where training and inference is performed using the method of [33] that can be used effectively on the large-scale data. We apply the DGP with two hidden layers and one output layer on all data-sets in our experiment.

- Isotonic regression (ISR): A simple non-parametric post-processing calibration method [19], which can recalibrate any regression algorithm similar to Platt scaling for classification. It is a separate auxiliary model to calibrate the probability output for a pre-train model and does not affect the original prediction of the model.

## B.2 Hyperparameters

Since the model structure is universal for all methods, we adjust the same optimal hyperparameters on the training data. Finally, we use the Adam [18] algorithm for the optimization with learning rate $10^{-4}$ and weight decay $10^{-3}$. For the kernel function of MMD, we use a mixture of six RBF kernels $k\left(x, x'\right) = \sum_{i=1}^{6} k_{\sigma_i}\left(x, x'\right)$ with $\sigma_i$ to be $\{1, 4, 8, 16, 32, 64\}$ in all experiments. For data preprocessing, we scale the data into the range [0,1] to avoid extreme values and improve the computation stability.

## B.3 Datasets

- Bike Sharing https://archive.ics.uci.edu/ml/datasets/bike+sharing+dataset: This dataset contains the hourly and daily count of rental bikes between years 2011 and 2012 in Capital bikeshare system with the corresponding weather and seasonal information at Washington, D.C., USA.

- Uber-pickup https://www.kaggle.com/yannisp/uber-pickups-enriched: This is a forked subset of the Uber Pickups in New York City from 01/01/2015 to 30/06/2015 from Kaggle, enriched with weather, borough, and holidays information.

- PM2.5 https://archive.ics.uci.edu/ml/datasets/Beijing+PM2.5+Data: This hourly data set contains the PM2.5 data of US Embassy in Beijing, time period is between Jan 1st, 2010 to Dec 31st, 2014. Missing data are filled by linear interpolation.

- Metro-traffic https://archive.ics.uci.edu/ml/datasets/Metro+Interstate+Traffic+Volum: Metro Interstate traffic volume dataset, hourly Interstate 94 Westbound traffic volume for MN DoT ATR station 301, roughly midway between Minneapolis and St Paul, MN from 2012-2018. Hourly weather features and holidays included for impacts on traffic volume.

- Air-quality https://archive.ics.uci.edu/ml/datasets/Air+Quality: The dataset contains 9358 instances of hourly averaged responses from an array of 5 metal oxide chemical sensors embedded in an Air Quality Chemical Multisensor Device. The device was located on the field in a significantly polluted area, at road level,within an Italian city. Data were recorded from March 2004 to February 2005 (one year)representing the longest freely available recordings of on field deployed air quality chemical sensor devices responses.

| Datasets | L | D | T |
|---|---|---|---|
| Uber-pickups | 29102 | 11 | 1 hour |
| Bike-sharing | 17389 | 16 | 1 hour |
| PM2.5 | 43824 | 13 | 1 hour |
| Metro-traffic | 48204 | 9 | 1 hour |
| Air Quality | 9358 | 15 | 1 hour |
| Power Plant | 9568 | 4 | nan |
| Protein Structure | 45730 | 9 | nan |
| Naval Propulsion | 11934 | 16 | nan |
| wine | 4898 | 12 | nan |

Table 3: The description of dataset used, where L is length of time-series or data size, D is number of variables, T is time interval among series.

# C   Metric Description

## C.1   The Metric of Prediction Precision

We also use RSE [20] and SMAPE [37] to quantify the prediction accuracy in addition to commonly used RMSE and $R^2$ for time-series forecasting tasks. Root relative squared error (RSE), that can be regarded as RMSE divided by the standard deviation of test data. Compared to RMSE, RSE is more readable because it can ignore the influence of data scale and it is able to recognize outlier prediction results. So lower RSE value is better. Where $y, \hat{y} \in \mathbb{R}^{n \times T}$ are ground truth and prediction value respectively.

$$RSE = \frac{\sqrt{\sum_{(i,t) \in \Omega_{\text{Test}}} (y_{it} - \hat{y}_{it})^2}}{\sqrt{\sum_{(i,t) \in \Omega_{\text{Test}}} (y_{it} - \mu(y))^2}} \tag{15}$$

Symmetric mean absolute percentage error (SMAPE or sMAPE) is an accuracy measure based on percentage errors. The absolute difference between $y_i$ and $\hat{y}_i$ is divided by half the sum of absolute values of the actual value $A_t$ and the forecast value $F_i$. The value of this calculation is summed for every fitted point $i$ and divided again by the number of fitted points $n$. Where $y_i, \hat{y}_i$ are the ground truth and prediction value respectively.

$$\text{SMAPE} = \frac{100\%}{n} \sum_{i=1}^{n} \frac{|\hat{y}_i - y_i|}{(|y_i| + |\hat{y}_i|)/2} \tag{16}$$

## C.2   The Metric of Calibration

Different from quantifying calibration in classification tasks, such as Brier Score [5], Reliability Diagrams [9] and Expected Calibration Error (ECE) [28], the calibration error is usually quantified by prediction intervals for regression tasks. In order to quantitatively evaluate the accuracy of predictive uncertainty, we use the numerical score of calibration error as an metric similar to the diagnostic tool proposed by [19]. Because the probability value is less than 1, in order to better distinguish the performance of calibration, here we use the absolute distance between expected confidence and observed confidence different from [19].

**Calibration error**. We designed two metrics, ECPE and MCPE to quantitatively evaluate our experiments. The expectation of coverage probability error (ECPE) of prediction intervals (PIs) is the absolute difference between true confidence and empirical coverage probability. Relatively, the maximum value of coverage probability error (MCPE) of prediction intervals(PIs) is the maximum distance.

$$ECPE = \frac{1}{n} \sum_{j=1}^{n} |P_j - \hat{P}_j| \tag{17}$$

$$MCPE = \max |P_j - \hat{P}_j| \tag{18}$$

Where $P_j$ is the expected confidence (i.e., the confidence level that we expect), and $\hat{P}_j$ is probability that prediction intervals cover the ground truth.

**Sharpness**. Another important aspect for evaluating calibration is the sharpness. We prefer prediction intervals as tight as possible while accurately covering the ground truth in regression tasks. We measure the sharpness using EPIW and MPIW. The expectation of prediction interval widths (EPIW) is averaged width of PIs, the maximum of prediction interval widths (MPIW) is the maximum width of PIs, reflecting the degree of uncertainty.

$$EPIW = \frac{1}{n} \sum_{j=1}^{n} \hat{Y}_{jup} - \hat{Y}_{jlow} \tag{19}$$

$$MPIW = \max(\hat{Y}_{jup} - \hat{Y}_{jlow}) \tag{20}$$

Where $\hat{Y}_{jup}, \hat{Y}_{jlow}$ are the upper and lower bounds of prediction intervals respectively.

Figure 5 and 7 shows the proportion that PIs covering ground truths at different confidence levels. The result of our proposed model is most close to the diagonal line, which indicates the best uncertainty calibration among all methods. Figure 6 shows the predictions and corresponding 95% prediction intervals. The intervals are visually sharp and accurately cover the ground truths.

(a) Dataset: Metro Traffic.

(b) Dataset: PM2.5.

Figure 5: Evaluating visually the quality of uncertainty by reliability diagrams. For each dataset, we plot the expected confidence vs observed confidence (empirical coverage probability) on the test data for compared baselines and proposed methods. It is obvious from the figure that observed confidence by our method is almost equal to the expected confidence.

(a) Dataset: Metro Traffic.

(b) Dataset: PM2.5.

Figure 6: Calibrated forecasting: Displayed prediction intervals (PIs) obtained at 95% confidence level by our proposed method in a time-series. As shown in the figure, the prediction intervals are also sharp while accurately covering the ground truth.

(a) Dataset: Wine.

(b) Dataset: Power Plant.

Figure 7: Evaluating visually the quality of uncertainty by reliability diagrams. For each dataset, we plot the expected confidence vs observed confidence (empirical coverage probability) on the test data for compared baselines and proposed methods. It is obvious from the figure that observed confidence by our method is almost equal to the expected confidence.

| Dataset | Metric | HNN | Deep-ens | MC NLL | ELL | DGP | proposed |
|---|---|---|---|---|---|---|---|
| Metro-traffic | EPIW | 786.42 | 788.03 | 1416.62 | 826.18 | 1155.69 | **776.05** |
| | MPIW | 2564.12 | 2569.37 | 4618.86 | 2693.74 | 3768.11 | **2530.30** |
| Bike-sharing | EPIW | 53.88 | 53.85 | 103.31 | 72.95 | 102.62 | 56.38 |
| | MPIW | 175.68 | 175.31 | 336.83 | 237.84 | 334.61 | 183.82 |
| Pickups | EPIW | 656.63 | 610.24 | 1748.87 | 827.60 | 847.00 | 625.57 |
| | MPIW | 2140.94 | 1989.67 | 5702.16 | 2698.37 | 2761.65 | 2039.67 |
| PM2.5 | EPIW | 90.20 | 82.97 | 125.07 | 71.89 | 114.40 | 87.82 |
| | MPIW | 294.09 | 270.51 | 407.87 | 234.41 | 373.01 | 286.35 |
| Air-quality | EPIW | 108.50 | 105.59 | 143.43 | 105.60 | 145.98 | **104.79** |
| | MPIW | 353.77 | 344.27 | 467.66 | 347.77 | 475.95 | **341.68** |

Table 4: The calibration sharpness of uncertainty evaluation for each method on different datasets. Our method produces relatively sharp prediction intervals, note that the smaller width of the prediction interval is not better without the guarantee of smaller calibration error. We prefer the prediction interval as tight as possible while accurately covering the ground truth.

| Dataset | Metric | MCD | HNN | Deep-ens | MC NLL | ELL | DGP | proposed |
|---|---|---|---|---|---|---|---|---|
| Metro-traffic | RMSE | 523.6 | 556.3 | 508.9 | 631.6 | 613.5 | 646.4 | 545.5 |
| | $R^2$ | 0.930 | 0.921 | 0.934 | 0.899 | 0.904 | 0.894 | 0.925 |
| | SMAPE | 15.76 | 15.68 | 14.90 | 21.21 | 18.41 | 20.82 | 17.47 |
| | RSE | 0.275 | 0.293 | 0.266 | 0.332 | 0.322 | 0.344 | 0.279 |
| Bike-sharing | RMSE | 38.86 | 40.71 | 37.60 | 89.57 | 52.50 | 55.39 | 37.93 |
| | $R^2$ | 0.912 | 0.904 | 0.918 | 0.536 | 0.841 | 0.823 | 0.917 |
| | SMAPE | 36.54 | 31.98 | 27.25 | 63.96 | 35.59 | 45.94 | 29.38 |
| | RSE | 0.318 | 0.339 | 0.302 | 0.968 | 0.459 | 0.481 | 0.310 |
| Pickups | RMSE | 350.3 | 359.8 | 336.4 | 526.8 | **325.9** | 440.3 | 346.9 |
| | $R^2$ | 0.967 | 0.965 | 0.969 | 0.925 | **0.971** | 0.944 | 0.967 |
| | SMAPE | 7.990 | 7.572 | 7.006 | 11.825 | **6.824** | 12.55 | 7.686 |
| | RSE | 0.189 | 0.194 | 0.181 | 0.295 | **0.176** | 0.249 | 0.185 |
| PM2.5 | RMSE | 70.95 | 58.81 | 60.24 | 66.77 | 61.09 | 61.44 | **57.43** |
| | $R^2$ | 0.154 | 0.264 | 0.389 | 0.250 | 0.372 | 0.365 | 0.298 |
| | SMAPE | 52.91 | 52.91 | 49.66 | 56.87 | 50.675 | 51.55 | 53.24 |
| | RSE | 1.111 | 1.083 | 1.290 | 1.930 | 1.254 | 1.469 | **1.080** |
| Air-quality | RMSE | 81.16 | 79.60 | 80.03 | 87.12 | 90.01 | 86.05 | 80.69 |
| | $R^2$ | 0.829 | 0.836 | 0.834 | 0.804 | 0.790 | 0.808 | 0.832 |
| | SMAPE | 26.97 | 24.13 | 24.60 | 28.03 | 27.37 | 30.97 | 24.88 |
| | RSE | 0.451 | 0.451 | 0.454 | 0.511 | 0.535 | 0.483 | 0.456 |

Table 5: The prediction precision of each method on different datasets. We report the RMSE, $R^2$, SMAPE and RSE for each of the cases, each row has the results of a specific method in a particular metric. Our proposed method achieves competitive results in prediction precision, almost outperforming the results of HNN in all metrics.