[Reviews · NeurIPS 2020]

Review 1

Summary and Contributions: Post-rebuttal: -------------------------------------------------------------------------------------------------- Thank you for the clarification. I decided to increase my score since I think the authors have provided very satisfactory answers to my concerns in the rebuttal. -------------------------------------------------------------------------------------------------- This paper proposes a calibration technique for regression models by matching (under maximum-mean discrepancy (MMD)) the model's predictive distribution to the true distribution, as represented by the empirical distribution. This distribution matching technique can be applied to any heteroscedastic regression network (e.g. the one used in deep ensembles Lakshminarayanan et al., NIPS 2017), and produce better calibration performance compared to various baselines in both forecasting and regression tasks.

Strengths: - The proposed method is very straightforward and can be applied to any heteroscedastic regression network (networks that output both the mean and variance of the Gaussian predictive distribution). - The authors provide theoretical insights on why calibration via a distribution matching is desirable in Theorem 1. - The proposed method yields significantly lower calibration errors compared to previous methods.

Weaknesses: - The method is only applicable to regression problems. (Or at least the authors only discussed the method in terms of regression.) - Theorem 1 is only applicable under well-specified models Q. In practice we are dealing with misspecified models, so I am not sure how useful the theory is in this case. - Guo et al. (ICML 2017) argue that miscalibration is due to the sheer size of modern NNs. However, the authors only validate the calibration performance of the proposed method on small networks (2 hidden layers with 256 units). - Lack of comparison with post-hoc calibration methods, such as temperature scaling.

Correctness: - The proposed method is sound and intuitive. I think the author did a good job of motivating it. - The authors claimed that the proposed method is "essentially different" from post-hoc calibration methods. I do not see why since one can apply the distribution matching procedure (Alg. 1, line 6-11) to any pre-trained heteroscedastic network. I think the paper would be much stronger if the authors argue that the proposed method is indeed a post-processing method since this means practitioners are not required to re-train their existing networks but can still enjoy the good calibration performance that the proposed method offers. - The claim in Theorem 1 and its proof appears to be correct. However, the proof could be better presented. E.g. the last two sentences in the proof do not read well.

Clarity: The paper is well written with some minor errors and vagueness: - In Eq. 7, there seems to be a clash of notation between y_i as an argument of L and as a variable inside the sum in the r.h.s. - Eqs. 10 and 11 describe a negative log-likelihood. However, the authors define L as a function of the data, instead of the parameters. - What does "...our method gets converged with one time iteration of the two stages..." in line 168 mean? It is unclear to me. - Line 173: What does "in a comprehensive way" mean here? - Fig. 2: Why is "sharpness" desirable? It should be discussed somewhere.

Relation to Prior Work: The authors did a good job on discussing how this work differs from previous works.

Reproducibility: Yes

Additional Feedback: - The paper would be much better in my opinion if the authors addressed the weaknesses I listed above, especially regarding the comparison with post-hoc methods and additional experiments with larger networks. - Additionally, the authors should compare with newer BNN baselines, such as SVDKL (Wilson et al, NIPS 2016), KFL (Ritter et al., ICLR 2018). - Please add error-bars to your results (Table 1 & 2; Fig. 3 & 4). - Table 1 & 2: Make the best RMSE values bold.


Review 2

Summary and Contributions: An uncertainty calibration method is proposed that uses MMD distribution matching for regression and time series forecast tasks. The idea behind the method is to minimize the MMD distance measure between the ground-truth labels and random samples from the predictive distribution. The technique allows learning calibrated predictive uncertainty at the distribution level. The proposed method is theoretically guaranteed to be well-calibrated in large data limit. Empirical results confirm that the proposed method can produce reliable predictive distribution with well-calibrated prediction intervals.

Strengths: The proposed technique is theoretically sound. Rigorous empirical evaluation confirms its utility in practice and competitiveness against alternative approaches.

Weaknesses: The proposed method is theoretically guaranteed to be well calibrated regressors given enough data. The paper could be stronger with some more insight into how to judge how much data is enough for a given problem.

Correctness: yes

Clarity: The paper is well written, clearly explains the proposed method giving the necessary background, discusses related work and provides comparison against alternative techniques.

Relation to Prior Work: yes

Reproducibility: Yes

Additional Feedback: I've read the author feedback, which addresses the question about practical use of the proposed method.


Review 3

Summary and Contributions: This paper considers the problem of probability calibration for regression tasks, and proposes a method based on the MMD to improve the level of calibration.

Strengths: The overall methodology is sounding. Given MMD's recent successes to distribution matching problems, it sure provides an open opportunity to improve regression calibration. The proposed method includes steps to check NLL and MMD simultaneously, which is also a valid option to optimise calibration.

Weaknesses: 1. The main issue of this paper is that it does not distinguish the concepts of quantile-level calibration and distribution-level calibration as in regression tasks. While the paper claims to work on distribution-level calibration, the definition introduced in section 2.1 is only quantile-level calibration, and is being quantile-calibrated is only a necessary condition for being distribution-calibrated (as discussed in ref.[34]). 2. The issue above further leads to a problem with the MMD calculation as described in eq.7. As the equation states, the proposed method only calculate a single MMD value for the entire dataset (e.g. with n data points). This indicates that the calculation only considers a single pair of distribution (P-Q). This hence contradicts the idea of distribution-level calibration. For instance, in distribution-level calibration, it assumes the model's predictions include a set of distributions (e.g. {Q_1, ..., Q_m}, where m <= n), and each of them will correspond to an unknown true distribution (e.g. Q_1 and P_1, ..., Q_m and P_m). Therefore, to ensure the matching between all these distributions, MMD needs to be calculated separately. Theorem 1 and eq.1 are therefore not concerning distribution-level calibration but built on a global quantile level. Even later in algorithm 1 it can be seen they only use a subset of data points during each iteration, it was mainly for SGD purposes, and not organising the subsets according to predicted distributions.

Correctness: As discussed above, the method and claims are valid for quantile-level distributions, but less accurate for distribution-level calibration.

Clarity: Yes, readers with related background should have a clear picture after a single read.

Relation to Prior Work: The paper covers and discusses a fair amount of related work; the only issue is that it leaves some ambiguities on quantile-level calibration and distribution-level calibration.

Reproducibility: Yes

Additional Feedback: While I am on the reject side now for reasons above, I still appreciate the efforts to create a regression calibration method using MMD. From the current theorem, methods and experiments, I can agree that this paper provides specific contributions towards quantile-level calibration, and leaves some open opportunities for future work on distribution calibration. So if the authors also agree they only work on quantile-level calibration for the current method and change their paper accordingly, I am open to increasing my score after the author response. ===========AFTER AUTHOR FEEDBACK===================== Since the authors clarify that they are working on the problem of quantile-level calibration, it solves my major concern. I therefore increase my score.


Review 4

Summary and Contributions: This paper proposes a method to generate a distribution matching strategy to get well-calibrated distributions that can output predictive uncertainties at all confidence levels. The main idea is to reduce the distance between the predicted uncertainty and the true probability uncertainty by minimizing the maximum mean discrepancy (MMD). The authors demonstrate the flexibility and effectiveness of the proposed method through extensive empirical evaluations.

Strengths: 1) The paper theoretically prove that the calibrated error of the proposed model asymptotically converges to zero then sample size is large enough. 2) the proposed MMD loss allows to learn the calibrated predictive uncertainty at the distribution level and can be generalized to confidence level without retraining. 3) The effectiveness of the method is evaluated extensively on several public datasets and show promising results comparing with the other state-of-the arts methods 4) The proposed method also shows higher computational efficiency (less computation time) among all the competing methods when model complexity becomes higher.

Weaknesses: The assumption in Gaussian likelihood maybe too restrictive, as noted by the authors, it'll be interesting to explore other base models in future work.

Correctness: This paper seems to be technically correct.

Clarity: This paper is well written and easy to follow.

Relation to Prior Work: This paper thoroughly discussed the difference of the proposed method from previous contributions.

Reproducibility: Yes

Additional Feedback: I have read the author's rebuttal others reviews and would like to keep my original score (an accept)

[Author Response · NeurIPS 2020]

We thank all reviewers for their valuable comments. We'll further improve in the final version. Below, we address the
detailed comments.

**To Reviewer 1:**

*Q1: Beyond regression tasks*: In this paper, we focus on regression tasks. As noted in the discussion part, we leave the
classification task as future work and point out the potential challenge to be solved.

*Q2: Model mis-specification*: Thanks. It's indeed an interesting topic to systematically investigate on the robustness of
our algorithm. We note that there is some recent work (e.g., [*1]) that studies the robustness of the MMD estimators,
which can be applied to our MMD-based calibration method.

[*1] Briol et al., Statistical inference for generative models with maximum mean discrepancy. arXiv:1906.05944.

*Q3. Results of larger networks*: Guo et al. (ICML 2017) argued that the miscalibration was due to the sheer size of
modern NNs and this conclusion was drawn from the image classification experiments. This conclusion is not suitable
for the regression tasks because the larger network may overfit. The size of our networks is close to the previous works
on regression problems ([11,16,18,20,30,35,38]). We'll make it clearer in the final version.

*Q4: Lack of comparison with post-hoc calibration methods*: Thanks. We have added the results of the related post-
hoc methods (isotonic regression), and the ECPE of isotonic regression is $0.032 \pm 0.002, 0.042 \pm 0.002, 0.011 \pm$
$0.000, 0.023 \pm 0.002, 0.030 \pm 0.001, 0.061 \pm 0.003, 0.014 \pm 0.006, 0.213 \pm 0.212$ and $0.083 \pm 0.006$ respectively for
Metro-traffic, Bike-sharing, Pickups, PM2.5, Air-quality, power-plant, protein-structure, naval-propulsion and wine.
We can see that the calibration errors of our method are significantly smaller than those of isotonic regression. The
updated results and error-bars will be included in the final version.

*Q5: Ambiguity of the argument "essentially different from post-processing methods"*: Thanks for clarifying and we
agree. Following the comment, We will change this argument to "our method also uses a post-processing procedure
but learns the calibration in the model level, which means practitioners are not required to retrain the model but can
enjoy the calibration performance".

*Q6: Clarity*: Thanks. We'll improve the clarity. In particular, we'll correct the minor errors in Eqn.7 and Eqn.10-11,
revise the corresponding part of Eqn.7 and add the model parameters in Eqn.10-11. We address the other points below:
1) *Line 168*: It means that our method is different from the alternating fashion [35], our method optimizes two loss
functions in turn.

2) *Line 173*: We mean that our distribution matching strategy produces the predictive distribution estimator (HNN) in
the model level (see Q5). We will make it more precise in the final version.

3) *Fig. 2 about "sharpness"*: We prefer prediction intervals as tight as possible while accurately covering the ground
truth in regression tasks. We measure the sharpness using the width of prediction intervals in this paper, which is
detailed in Appendix C. Sharpness was previously used in [18, 35]. We will add an explanation in the main text.

*Q7: newer BNN baselines*: We have compared with DeepGP (Hugh et al, NIPS2017), which is a strong Bayesian
baseline and can produce well-calibrated predictive uncertainties. We also add a BNN baseline method named fpovi
[*2], whose ECPE is $0.065 \pm 0.001, 0.132 \pm 0.002, 0.173 \pm 0.001, 0.261 \pm 0.000$ and $0.276 \pm 0.001$ for Metro-traffic,
Bike-sharing, Pickups, PM2.5 and Air-quality. The calibration error of our method is significantly smaller.

[*2] Ziyu et al. Function Space Particle Optimization for Bayesian Neural Networks. ICLR 2019.

**To Reviewer 2:** As for the amount of data is enough for a given problem, [12] shows that MMD has performance
guarantees at finite sample sizes, based on uniform convergence bounds. For the results of our method, regardless of
whether or not $p = q$, the empirical MMD converges in probability at rate $O((m + n)^{-\frac{1}{2}})$ to its population value,
where $m$ and $n$ respectively represent the number of samples sampled from $P$ and $Q$. And in practical applications,
you can judge whether the data is sufficient according to MMD error. We will add the related discussion in the final
version.

**To Reviewer 3:** Thanks for clarifying the concept of the distribution-level calibration and sorry for the confusion. The
"distribution-level" in our submission means that we learn the calibrated predictive distribution $Q$ by minimizing the
kernel embedding measure. Yet, this claim does not mean to imply the pre-existing "distribution calibration" in [34].
According to [34], we agree that our method is for the global quantile-level calibration. To avoid ambiguity here, we
will remove the words "distribution-level" in the final version and more precisely say that "we learn the calibrated
predictive distribution $Q$ by minimizing the kernel embedding measure".

**To Reviewer 4:** Thanks for the positive review.

[Meta-Review · NeurIPS 2020]

This paper was reviewed by 4 reviewers with significant expertise on the topic. All 4 were clearly in favor of accepting the paper and I agree with their opinion and recommend that it be accepted. I strongly encourage the authors to take into account the reviewer suggestions when revising the paper, particularly R1 and R3 who provided a number of detailed points for improvement.